# iTRAQ-Based Proteomics Analysis of Response to *Solanum tuberosum* Leaves Treated with the Plant Phytotoxin Thaxtomin A

**DOI:** 10.3390/ijms222112036

**Published:** 2021-11-07

**Authors:** Lu Liu, Liaoyang Hao, Ning Liu, Yonglong Zhao, Naiqin Zhong, Pan Zhao

**Affiliations:** 1College of Life Science, University of Chinese Academy of Sciences, Beijing 100049, China; liulu183@mails.ucas.ac.cn; 2State Key Laboratory of Plant Genomics, Institute of Microbiology, Chinese Academy of Sciences, Beijing 100101, China; 3School of Agriculture, Ningxia University, Yinchuan 750021, China; xiongba_732345670@163.com (L.H.); ylzhao97101@163.com (Y.Z.); 4National Engineering Research Center for Vegetables (Beijing Vegetable Research Center), Beijing Academy of Agricultural and Forestry Sciences, Beijing 100097, China; liuning@nercv.org; 5Engineering Laboratory for Advanced Microbial Technology of Agriculture, Chinese Academy of Sciences, Beijing 100101, China; 6The Enterprise Key Laboratory of Advanced Technology for Potato Fertilizer and Pesticide, Hulunbuir 021000, China

**Keywords:** *Solanum tuberosum*, common scab, thaxtomin A, proteomic analysis, iTRAQ

## Abstract

Thaxtomin A (TA) is a phytotoxin secreted by *Streptomyces scabies* that causes common scab in potatoes. However, the mechanism of potato proteomic changes in response to TA is barely known. In this study, the proteomic changes in potato leaves treated with TA were determined using the Isobaric Tags for Relative and Absolute Quantitation (iTRAQ) technique. A total of 693 proteins were considered as differentially expressed proteins (DEPs) following a comparison of leaves treated with TA and sterile water (as a control). Among the identified DEPs, 460 and 233 were upregulated and downregulated, respectively. Based on Gene Ontology (GO) and Kyoto Encyclopedia of Genes and Genomes (KEGG) analyses, many DEPs were found to be involved in defense and stress responses. Most DEPs were grouped in carbohydrate metabolism, amino acid metabolism, energy metabolism, and secondary metabolism including oxidation–reduction process, response to stress, plant–pathogen interaction, and plant hormone signal transduction. In this study, we analyzed the changes in proteins to elucidate the mechanism of potato response to TA, and we provided a molecular basis to further study the interaction between plant and TA. These results also offer the option for potato breeding through analysis of the resistant common scab.

## 1. Introduction

The potato (*Solanum tuberosum*) is the third most important food crop in the world after wheat and rice, and it is widely grown on all continents except for Antarctica [1]. Due to its high yield and strong adaptability, it is used to ensure food security [2]. However, potato common scab (CS) is one of the main soil-borne diseases that is caused by pathogenic *Streptomyces* spp., and it has occurred in all growing areas of the world in recent years [3]. The symptoms of CS include scab-like superficial, raised, and/or pitted lesions that severely reduce tuber quality and market value [4,5,6]. The symptoms are induced by thaxtomins that are secreted by pathogenic *Streptomyces* spp. [7].

Thaxtomins are nitro-aromatic compounds that were originally described by King and Lawrence [8,9]. The predominant form is thaxtomin A (TA), and it is produced by *Streptomyces scabies* [7,10]. TA is synthesized by a nonribosomal peptide synthetase (NRPS) that is encoded by the *txtA* and *txtB* genes [11]. NRPS is responsible for the biosynthesis of the N-methylated cyclic dipeptide backbone of this toxin [6]. Post-cyclization hydroxylation is conducted by a cytochrome P450 monooxygenase that is encoded by *txtC* [12]. The nitration of the dipeptide is carried out by a nitric oxide synthase (NOS) encoded by *nos/txtD* in the vicinity of *txtAB* [13]. Then, site-specific nitration is catalyzed by a cytochrome P450 encoded by *txtE* [14]. It was predicted that an additional gene (*txtH*) in the thaxtomin biosynthetic gene cluster of *S. scabies* encodes a member of the MbtH-like protein superfamily [15]. These biosynthetic genes of thaxtomin phytotoxins (txt) are clustered and located on a pathogenicity island (PAI) in the genome of *S. scabies*, *S. turgidiscabies*, *S. acidiscabies*, and *S. ipomoeae* [16,17,18]. An AraC/XylS-type transcriptional regulator encoded by *txtR* controls the toxin biosynthesis [19,20].

As an inhibitor of cellulose synthesis, TA induces atypical programmed cell death (PCD); affects Ca^2+^, H^+^, and K^+^ flux; increases polysaccharide deposition; and produces tissue necrosis [21,22,23,24]. However, the involved molecular mechanisms are not very clearly defined, as only the cellooligosaccharide-mediated pathway has been detailed [19,25,26,27]. Thus, proteomics works have been used to investigate the mechanism and assess the *S. scabies* pathogenesis elicited under different conditions. Two-dimensional polyacrylamide gel electrophoresis (PAGE) was used to detect proteins induced in *S. scabies* by potato suberin, and the metabolic pathways, proteins involved in secondary metabolism, and morphological differentiation were analyzed [28]. The comparative extracellular proteomic analysis of *S. scabies* showed that the multiple secretion of proteins in the twin-arginine translocation (Tat) pathway was required for full virulence [29].

A comparative secretome showed that the presence of suberin in an *S. scabies* growth medium induced the production of a wide variety of glycosyl hydrolases and identified enzymes involved in the degradation of suberin that occurred during carbohydrate and lipid metabolism [30]. Moreover, comparative proteomic analysis was used to study *S. scabies* grown in the absence or presence of cellobiose, revealing the role of CebR as a regulator of the virulence of *S. scabies* [31].

As powerful tools, proteomic technologies can also discern global protein expression changes during plant responses to biotic and abiotic stresses at the cellular and molecular levels. The Isobaric Tags for Relative and Absolute Quantitation (iTRAQ) technology has been used in proteomics studies due to its high efficiency and accuracy in identifying differentially expressed proteins (DEPs) [32,33,34]. This technology is used to analyze proteomics in response to different biotic or abiotic stresses, including heat stress [35], salt stress [36], drought tolerance [37], rice response to *Magnaporthe oryzae* [38], defense response in cotton triggered by *Rhizoctonia solani* [39], and *Arabidopsis* response to *Verticillium dahliae* [40].

In this study, we used iTRAQ-based quantitative proteomics to analyze the change in proteins in potato leaves treated with TA stress. The authenticity and accuracy of the results of the proteomic analysis were confirmed by quantitative reverse transcription-polymerase chain reaction (qRT-PCR). A number of DEPs were associated with defense response, amino acid metabolism, carbohydrate metabolism, and other functions. We also detected the antioxidant system response after TA treatment. Our results showed the response by the potato leaves to TA stress and provide a theoretical basis for understanding TA-resistance mechanisms.

## 2. Results

### 2.1. Effects of TA Treatment on Potato Leaves

It has been known that TA can trigger PCD in the host [23]. To confirm the phenotype in the potato, the *Solanum tuberosum* leaves were treated with different concentrations of TA. After treating potato leaves for 24 h with TA (10, 50, and 100 μM), control leaves and TA-treated leaves were collected. The effect of concentration changes in response to TA was tested using trypan blue staining, which stained the dead cells to dark blue while the living cells remained unstained. As shown in Figure 1a, necrotic spots began to appear on leaves from 10 μM TA onward. With the increase in the concentrations of TA, the number of necrotic spots on the leaves increased. When the concentration of TA reached 100 μM, the potato leaves were totally covered with necrotic spots, so it was concluded that 10 μM TA could induce the physiological response. The results were also confirmed by the phenotype of the potato plants (Figure 1b).

### 2.2. Quantitative Identification of Proteins

To understand the molecular pathways that the host applied in response to the TA treatment, potato leaves treated with sterile water (as control) and 10 μM TA were subjected to iTRAQ analysis. After merging data, a total of 33,881 peptides and 11,408 proteins were identified in these two treatments. After protein grouping, the identified peptides/proteins were classified into 5942 groups (Figure 2a). The distribution of protein molecular weight is shown in Figure 2b. Notably, the molecular weight of 1047 proteins fit in the 1–21 kDa group, 2118 proteins were classified into the 21–41 kDa group, 1481 proteins were classified into the 41–61 kDa group, 653 proteins were classified into the 61–81 kDa group, and others were classified into the >81 kDa group (Figure 2b).

At the next step, all proteins were aligned to the database. Then, 5835 proteins were functionally annotated. DEPs were defined as fold change >1.2 or fold change <0.83 and *p*-value <0.05. The iTRAQ analysis showed that there were 693 DEPs identified when comparing the TA treatment and the control set. Of the 693 DEPS, 460 DEPs were upregulated and 233 DEPs were downregulated in the TA treatment compared to the control (Figure 3).

### 2.3. Analysis of DEPs

Gene Ontology (GO) functional annotation was applied to categorize the 693 DEPs (Figure 4). The results of all 46 GO functional annotation statistics showed that 20 GO terms belonged to the biological process (BP) category, 14 GO terms belonged to the cellular component (CC) category, and 12 GO terms were classified into the molecular function (MF) category. Then, GO functional enrichment analysis was carried out (Appendix A). Compared to the control, DEPs in the BP category were mainly enriched in the biological (392 DEPs), single-organism (204 DEPs), single-organism metabolic (166 DEPs), and oxidation–reduction processes (99 DEPs). In the CC category, DEPs were mainly enriched in the membrane (74 DEPs), chloroplast (59 DEPs), and plastid (59 DEPs) parts. In the MF category, DEPs were mainly enriched in catalytic activity (278 DEPs), oxidoreductase activity (96 DEPs), and metal-ion binding (89 DEPs). In additional, we found some GO terms that contained a few proteins also worth consideration given their function in resistant pathways: response to stimulus (65 DEPs), response to stress (53 DEPs), defense response (18 DEPs), detoxification (17 DEPs), response to biotic stimulus (13 DEPs), secondary metabolic process (6 DEPs), response to bacterium (6 DEPs) in the BP category; photosynthetic membrane (37 DEPs), Golgi apparatus (9 DEPs), and photosystem I reaction center (5 DEPs) in the CC category; calcium ion binding (18), antioxidant activity (17 DEPs), peroxidase activity (14 DEPs), and oxidoreductase activity (acting on NAD(P)H, 11 DEPs) in the MF category.

Next, a Clusters of Orthologous Groups (COG) analysis was conducted. The results showed that all DEPs could be grouped into 21 functional classes (Figure 5). The largest group was ”posttranslational modification, protein turnover, chaperones“, which included 54 DEPs and belonged to the ”cellular processes and signaling“ type. Groups 2–4 were classified as ”metabolism“ types—”carbohydrate transport and metabolism” (42 DEPs), ”energy production and conversion“ (32 DEPs), and ”amino acid transport“ (27 DEPs), respectively. The fifth largest group was ”metabolism and lipid transport“ (27 DEPs), which was classified as “poorly characterized”.

In addition to the GO-based annotation, we applied the Kyoto Encyclopedia of Genes and Genomes (KEGG) annotation for the DEPs. The analysis showed that the corresponding pathways could be divided into 6 primary categories (metabolism, genetic information processing, environmental information processing, cellular processes, organismal systems, and human diseases), including 19 s categories (Figure 6). The most abundant category was carbohydrate metabolism (58 DEPs). The second to fifth categories were global and overview maps (49 DEPs); amino acid metabolism (46 DEPs); energy metabolism (42 DEPs); and folding, sorting and degradation (41 DEPs), respectively. Then, KEGG enrichment analysis was conducted, and DEPs were enriched into 101 different pathways (Appendix A). The top five pathways of enriched proteins were protein processing in endoplasmic reticulum (33 DEPs), phenylpropanoid biosynthesis (26 DEPs), carbon metabolism (26 DEPs), the biosynthesis of amino acids (24), and glutathione metabolism (18 DEPs). Although some pathways only include a few proteins—e.g., the phenylalanine, tyrosine, and tryptophan biosynthesis pathway (11 DEPs); plant–pathogen interaction (11 DEPs); peroxisome (8 DEPs); plant MAPK signaling pathway (7 DEPs); tricarboxylic acid (TCA) cycle (5 DEPs); plant hormone signal transduction (3 DEPs); tryptophan metabolism (2 DEPs); and other autophagy (1 DEP)—their function cannot be ignored. 

Proteins perform various functions and usually interact with other proteins, so protein–protein interactions play key roles in the physiological process. We found 30 interaction groups in which the combined score was the highest; these were used to make the network diagram (Appendix A). In the diagram, each node represents a protein, and the lines between nodes represent the interactions between the two proteins. The size of the node represents the number of the interacting proteins. The bigger the node is, the more proteins that interact with it. In this diagram, PGSC0003DMP400017553 has five interaction proteins; PGSC0003DMP400037659, PGSC0003DMP400003002, PGSC0003DMP400029972 each have three interaction proteins. PGSC0003DMP400017553 is an upregulated protein that is annotated HSP90. This protein can respond to stress and take part in plant–pathogen interaction pathway. This analysis can provide valuable candidate proteins, such as PGSC0003DMP400017553, for further functional research.

### 2.4. Validation of DEPs by qRT-PCR

qRT-PCR was used to assess the correlation between gene expression level and protein abundance (Figure 7). We randomly selected 10 DEPs, including two no-changed proteins (PGSC0003DMP400034650 and PGSC0003DMP400003405), five upregulated proteins (PGSC0003DMP400002693, PGSC0003DMP400025999, PGSC0003DMP400036207, PGSC0003DMP400046968, and PGSC0003DMP400022299) and three downregulated proteins (PGSC0003DMP400000965, PGSC0003DMP400026922, and PGSC0003DMP400041818). The results of qRT-PCR showed consistency in transcription and protein abundance, as well as verifying the reliability of the identification of DEPs. PGSC0003DMP400002693, PGSC0003DMP400025999, and PGSC0003DMP400022299 clearly showed increases in transcription level rather than protein abundance.

### 2.5. Transcriptional Level Analysis of Selected DEPs

We found several DEPs involved in response to “oxidative stress”, and these DEPs have close relationships with plant defense. To verify the correlation between the transcripts and proteins of those DEPs after being treated with TA, nine genes related to oxidative stress response were checked by qRT-PCR (Figure 8). With increasing treatment time, the expression of those genes was significantly upregulated. The expression of a peroxidase (PGSC0003DMP400021162) reached its peak at 3 hpi, while the expression of another peroxidase (PGSC0003DMP400059654) reached its peak at 12 hpi. The expression of a superoxide dismutase (SOD) (PGSC0003DMP400000820) reached its peak at 6 hpi. The expression of two defense response genes (PGSC0003DMP400002693, PGSC0003DMP400053803) reached its peak at 3 hpi. The expression of a defense response gene (PGSC0003DMP400062364) reached its peak at 6 hpi. The expression of two defense response genes (PGSC0003DMP400003634 and PGSC0003DMP400051894) reached its peak at 12 hpi. The expression of a defense response gene (PGSC0003DMP400002803) reached its peak at 24 hpi.

### 2.6. Detection of Antioxidant System Response after TA Treatment

The data suggested that more DEPs were related to “oxidative stress” were upregulated. This may induce high (reactive oxygen species) ROS and triggers PCD. To confirm whether the ROS level was changed after TA treatment, leaves were tested by 3,3′-diaminobenzidine (DAB) staining. Compared to the control, leaves treated with 10 μM TA displayed dark brown spots (Figure 9a), which indicated that TA could significantly induce hydrogen peroxide (H_2_O_2_) accumulation. This result was confirmed with gene expression by qRT-PCR. To test the relationship between increased H_2_O_2_ and ROS-scavenging enzymes, we examined the activities of catalase (CAT) and peroxidase (POD). The activities of CAT and POD were significantly enhanced at 3 h. The activity peak of POD appeared at 12 h, and the activity peak of CAT appeared at 6 h (Figure 9b).

## 3. Discussion

Plants possess a defense system that consists of a sophisticated network called the innate immune system [41]. This system is triggered by invasion signals in two-tiered defense approaches [42,43]. Pathogen- and microbe-associated molecular pattern (PAMP/MAMP)-triggered immunity (PTI) is the first line of defense, and effector-triggered immunity (ETI) is the second. Both types of defense share immune responses, including the accumulation of PR proteins, ROS, callose, and secondary metabolites such as terpenes and tryptophan-derived metabolites [44,45]. PTI includes distinct physiological responses such as ROS, signal transduction, MAPK cascades for phosphorylation, callose deposition, PCD, hypersensitive response (HR), and defense hormones [46,47,48]. In *A. thaliana* cell cultures, TA can induce atypical PCD. PCD is associated with rapid ROS accumulation when a plant recognizes the pathogens [49]. ROS are important for defense signaling in response to pathogen attack [50,51]. The production of O^2·−^ and H_2_O_2_ in ROS is known to be involved in plant defense mechanisms [52,53,54] Plants control redox levels for proper cellular function with enzymes, including SOD, glutathione peroxidase (GPX), POD, and CAT [55,56]. In this study, we found that the TA treatment could mimic the invasion of *S**. scabies* in the *S. tuberosum*. ROS accumulation and necrotic spots were observed on the leaves once the plants were treated with 10 μM TA. Along with these observations, the proteomic analysis confirmed that the expression of several proteins involved in defense-related pathways were altered. Our data suggested that plant phytotoxins, such as TA, play critical roles in inducing host immune responses.

ROS are important molecules in regulating host immune responses. ROS signaling plays a dual role in mediating host defense. At low concentrations, ROS act as signal molecules that spread defense signaling to adjacent tissues, and high concentrations of ROS can trigger PCD. In our antioxidant system test, after treatment with TA, ROS produced by leaves were detected by the DAB test (Figure 9a). The ROS participating in stress signaling can be transduced by mitogen-activated protein kinases (MAPKs) [57,58]. MAPKs can act upstream of the oxidative burst during ozone treatment or work downstream in ROS-dependent cell death events [49]. In our study, eight DEPs were grouped in the MAPK signaling pathway for plants, and six DEPs were upregulated, with PGSC0003DMP400003634 (FC = 1.99) being the most significant DEP. Moreover, most of the proteins related to photosynthesis synthesis were downregulated, such as PGSC0003DMP400020951 (photosynthesis), PGSC0003DMP400046904 (chloroplast), and PGSC0003DMP400012373 (oxidative phosphorylation). We speculated that when ROS burst in potato leaves, the MAPK pathway increases the speed of the transduced signal and subsequently induces ROS-dependent cell death. We also tested the enzymatic activities of CAT and POD, and the activities of these two enzymes were significantly increased (Figure 9). This may indicate that when *S. scabies* invades potato plants and secretes TA, the plants produce ROS in response to defend against the harmful events and enhance the activities of POD and CAT to regulate the ROS. In our study, 12 DEPs annotated as peroxisomes were all upregulated, and the most upregulated DEP was PGSC0003DMP400053803 (FC = 1.46). In the 19 DEPs annotated as peroxidase, only 2 were downregulated and the most upregulated DEP was PGSC0003DMP400021162 (FC = 4.10). We supposed that after TA treatment, the potato leaves synthesized ROS to defend against stress and simultaneously regulated the expression of related enzymes to scavenge the ROS, which helped to maintain the ROS homeostasis.

The generation of ROS in a cell is a consequence of electron leakage during photosynthesis and respiration [59]. Our iTRAQ results indicated that there were 42 DEPs involved in energy metabolism from the KEGG annotation. In the KEGG enrichment analysis, the energy metabolism pathways include carbon metabolism, amino acid metabolism, carbon fixation in photosynthetic organisms, the TCA cycle, and the biosynthesis of amino acids pathway. These results imply that after TA treatment, plants change their carbon metabolism and amino acid metabolism to trigger cell wall repair and also produce ROS to eliminate the harmful effects of TA. Then, antioxidant systems moderate the overproduction of ROS and oxidative stress. There is a connection between callose deposition and ROS. Callose is a defensive polysaccharide that accumulates to form papilla that prevents pathogens from entering, thus acting as a marker of activated defense responses [60]. O^2-^ and ^1^O_2_ induce strong callose accumulation against *B. cinerea* in both *Arabidopsis* and tomato plants [61,62]. ROS may partially inhibit cell death by altering the cellular redox environment and reinforcing the cell wall to prevent or compensate for damages induced by TA [63]. This idea is consistent with our results that showed that ROS participate in the plant response to stress conditions to activate plant defenses.

TA can cause plant cell hypertrophy and cell death in field-infected potatoes [9], cause cellular collapse and tissue necrosis [64], specifically inhibit glucose incorporation into cellulosic tissue in the cell wall fraction of *Arabidopsis thaliana* seedlings [65], and inhibit root growth and cause root swelling [66,67]. TA also reduces crystalline cellulose and increases pectin and hemicellulose in the cell walls of *A. thaliana* seedlings. The expression of genes associated with pectin metabolism and the cell wall also can be changed [68].

KEGG annotation indicated that there were 58 DEPs involved in carbohydrate metabolism after TA treatment. Carbon metabolism plays a crucial role in plant stress response and maintains the supply of nutrients and energy. Carbon metabolism provides carbon skeletons for plant growth of primary metabolism or defense needs of secondary metabolism [69]. Plant defense should rely upon the energy drain from growth to product defensive metabolite. KEGG enrichment analysis showed that 26 DEPs were involved in carbon metabolism, 15 of which were upregulated, and all 5 DEPs involved in the TCA cycle were also upregulated. In carbon metabolism, the most upregulated DEP is PGSC0003DMP400003446 (FC = 2.087). This is a phosphogluconate dehydrogenase (PGD) that catalyzes the oxidative decarboxylation of 6-phosphogluconate, along with the concomitant reduction of NADP to NADPH. The KEGG analysis showed that the carbon metabolism DEPs were more upregulated than downregulated, which indicated that plant cellulose synthesis was enhanced. This may indicate that when *S. scabies* invades plants and produces TA, the plant produces carbohydrates such as starch, glucans, and xylan to decrease the harmful effects of TA. As previously reported, carbohydrates affect the production of TA by pathogenic *Streptomycetes*, and high levels of xylans and glucans can induce TA production [70]. TA interferes with carbon metabolism, and this suggests that when *S. scabies* invades a plant, the plant adjusts its carbon metabolism to change its cellulose synthesis in order to inhibit the production of TA.

Our data also suggested that the regulated expression of some host proteins may be involved in inhibiting TA biosynthesis. The KEGG annotation showed that there were 46 DEPs involved in amino acid metabolism. Amino acid metabolism is tightly linked to energy and carbohydrate metabolism [71]. Lysine accumulation affects starch synthesis [72], and glycolysis and the TCA cycle are regulated by alanine aminotransferase during hypoxia [73]. Additionally, alanine aminotransferase can regulate carbon and nitrogen metabolism [74]. Under an energy deficiency, lysine and isoleucine catabolism directly participate in the TCA cycle [75]. The KEGG enrichment analysis showed that there were 24 DEPs involved in the biosynthesis of amino acids pathway; other DEPs were involved in amino acid metabolism. Tryptophan or the methylation of tryptophan and aromatic amino acids (tyrosine and phenylalanine) can inhibit TA biosynthesis [76]. L-Tryptophan nitration and phenylalanyl hydroxylation are catalyzed by cytochrome P450 monooxygenase during the synthesis of TA [14]. We found that 2 DEPs involved in tryptophan metabolism; 10 DEPs involved in phenylalanine metabolism; 11 DEPs involved in phenylalanine, tyrosine, and tryptophan biosynthesis; 5 DEPs involved in tyrosine metabolism; and total DEPs were all upregulated except for one protein (PGSC0003DMP400022208, FC= 0.804) in phenylalanine, tyrosine, and tryptophan biosynthesis. 

Amino acid metabolism is also connected with the plant stress response. Lysine is catabolized through the saccharopine pathway, which has been shown to plays a role in abiotic and biotic stress responses [77,78,79]. Proline plays a major role in abiotic stress responses in developing maize seeds, and its role is similar to that of pipecolate in *Arabidopsis* and *Brassica napus* [77]. Pipecolate was also found to be significantly increased in *Arabidopsis* infected with pathogenic bacteria, playing a role in plant defense responses [78]. Homoserine and threonine are derived from the aspartate pathway, and their accumulation increases plant immunity to oomycete pathogens [80]. We speculated that, for a plant to overcome harmful effects after TA invasion, related amino acid synthesis pathways must be activated and detoxification protein abundance must increase. We identified 46 DEPs including PGSC0003DMP400055404 (FC = 2.53), PGSC0003DMP400022299 (FC = 3.26), and PGSC0003DMP400009021 (FC = 2.49). Except for detoxification, other biological processes such as hydrogen peroxide catabolic response to oxidative stress and the oxidation–reduction process were included. By regulating the amino acid and carbohydrate metabolism, the resistance proteins participate in multiple pathways and then alleviate the damage caused by TA. The important proteins in the protein interaction network play their function in related pathways. For example, PGSC0003DMP400037659 (FC = 0.79) participates in photosynthesis, and PGSC0003DMP400003002 (FC = 1.45) and PGSC0003DMP400029972 (FC = 0.72) participate in translation. The protein interaction analysis showed that the top protein was PGSC0003DMP400017553 (FC = 1.91) was HSP90, which can respond to stress and interact with PGSC0003DMP400050363 (FC = 1.29) (which is a plant–pathogen interaction protein), PGSC0003DMP400025062 (FC = 1.87) (which is a phagosome protein), and PGSC0003DMP400024824 (FC = 1.29) (which is a defense response protein that participates in the protein interaction).

In our study, we identified 693 DEPs from TA-treated leaves with the iTRAQ technique. Based on the iTRAQ analysis, the DEPs were categorized into different pathways including carbohydrate metabolism, amino acid metabolism, energy metabolism, oxidation–reduction process, response to stress, and the secondary metabolic process. The proteomic results were confirmed by qRT-PCR, DAB stain, and enzymatic activity tests. Our results showed a high correlation between transcripts and proteins, including ROS response, disease, defense response, PR proteins, and secondary metabolites; additionally, the gene transcription level was more significant than protein abundance (Figure 7 and Figure 8).

TA secreted by *S. scabies* triggered the plant immune response via changes in metabolism. TA can be used to test for resistance to CS in potato breeding [81,82,83]. Our results contribute to the understanding of the response mechanisms of potato plants to TA and provide a theoretical basis for potato breeding.

## 4. Materials and Methods

### 4.1. Plant Materials

The *S. tuberosum* cultivar “Shepody” and *S. tuberosum* group Phureja DM1–3 were micropropagated on a Murashige and Skoog (MS) medium containing 30 g/L of sucrose and 0.8% agar [84]. The pH of the media was adjusted to 5.8. Two weeks later, potato seedlings that had same node segments and similar lengths were transplanted into pots containing nutrient soil (soil:vermiculite =3:1) and then cultured at 23 °C with a 16 h light/8 h dark photoperiod and 70 μmol m^−2^ s^−1^ photon flux density provided by fluorescent lamps. Next, 4-week-old potato seedlings were transported to pots, and plants were subjected to treatment with water and 10, 50, and 100 μM TA. Three biological replicates were used for each treatment.

### 4.2. Thaxtomin A Preparation and Treatment

The methods of TA extraction, purification, and quantification from *S. scabies* were performed as previously described [7]. Briefly, *S. scabies* spores (1 × 10^6^ CFU/mL) were cultured overnight in 5 mL of an oat bran liquid medium at 28 °C on an orbital shaker at 200 rpm. Overnight cultures were transferred to 1 L of an oat bran liquid medium and incubated at 28 °C on an orbital shaker at 200 rpm for 7 days. After the centrifugation of the cultures, the supernatants were collected and then sterile-filtered with 0.22 μm filters. TA was extracted from the supernatants using an equal volume of ethyl acetate, which was subsequently removed by vacuum-rotary evaporation. The obtained yellow product, which was TA, was redissolved in 1–2 mL of methanol, lyophilized, and stored. TA was quantified by high-performance liquid chromatography (HPLC, Agilent Technologies, Santa Clara, CA, USA).

Solutions of different concentrations (10, 50, and 100 μM) were created by dissolving purified TA in distilled water. A TA solution containing 0.05% Tween-20 was sprayed onto potato leaves until runoff occurred, and water-treated leaves were used as a control. A moisturizing transparent high cover was used to protect the pots for 24 h to prevent evaporation. The necrotic spots in the leaves were observed with trypan blue staining [85]. Plants treated with 10 μM TA were prepared for protein extraction because this concentration of TA effectively induced a leaf response.

For the iTRAQ or qRT-PCR analysis, leaves of the *S. tuberosum* group Phureja DM1–3 from five different plants were pooled together as one sample at 24 h after TA treatment, immediately flash-frozen in liquid nitrogen, and then stored at −80 °C. Three biological replicates were used for each treatment.

### 4.3. Protein Extraction and Quantification

The total proteins were extracted from control and TA-treated leaves according to a previously published method with minor modifications [39,40]. Briefly, 0.5 g samples were rapidly ground in liquid nitrogen and suspended in tripled volumes of borax/PVPP/phenol (BPP) [86]. Samples were vortexed at 4 °C for 10 min. Then, an equal volume of Tris-saturated phenol (pH ≥ 7.8) was added, followed by vortexing at 4 °C for 10 min and centrifuging for 10 min (4 °C, 12,000× *g*). The supernatants were extracted by equal volumes of BPP. The collected supernatants were mixed with 5× volumes of a supersaturated sulfuric acid ammonium methanol solution (AM precipitator) and precipitated overnight at −20 °C. After centrifuging for 20 min (12,000× *g*, 4 °C), the supernatant was discarded. The pellet was mixed with ice-cold acetone and centrifuged for 5 min (12,000× *g*, 4 °C). Protein pellets were resuspended in 8 M urea containing sodium dodecyl sulfate (SDS, 1%) and protease inhibitor cocktail (1 X concentration). After centrifuging at 12,000× *g* and 4 °C for 20 min, the supernatants were quantified with a Pierce™ BCA Protein Assay Kit following the manufacturer’s instructions (Thermo Fisher Scientific, Waltham, MA, USA).

### 4.4. Protein Digestion, iTRAQ Labeling, and Liquid Chromatography–Tandem Mass Spectrometry (LC–MS/MS)

Two groups of proteins were assayed using iTRAQ by Shanghai Majorbio Bio-pharm Technology Co., Ltd. (Shanghai, China, http://www.majorbio.com/; accessed on 18 January 2019). The procedure used for the iTRAQ analyses is as follows.

First, 100 μg of proteins from each group were reduced in 10 mM Tris (2-carboxyethyl) phosphine (TCEP, Thermo Fisher Scientific, Waltham, MA, USA) at 37 °C for 60 min and then alkylated using iodoacetamide (40 mM) for 30 min in the dark at room temperature. Pre-chilled 6× acetone was added at −20 °C for 4 h and then centrifuged at 10,000× *g* for 20 min. The treated proteins were dissolved in 100 mM triethylammonium bicarbonate (TEAB) and digested overnight at 37 °C with trypsin (Promega, Madison, WI, USA) using a protein:enzyme ratio of 50:1 (*w*/*w*). Samples were then labeled following the manufacturer’s instructions for the iTRAQ Reagents 8-Plex kit (AB Sciex, Framingham, MA, USA). The control samples were labeled with 113, 115, and 117 tags, and the TA-treated samples were labeled with 118, 119, and 121 tags.

The samples were dissolved in buffer A (2% acetonitrile in aqueous ammonia; pH 10.0) and fractionated using a Thermo Scientific Vanquish Flex Ultra-High Performance Liquid Chromatography (UHPLC) system equipped with an Acquity UPLC BEH C18 column (2.1 × 150 mm, 1.7 μm, Waters, Milford, MA, USA). The peptides were eluted using a linear gradient of buffer B (80 vol% acetonitrile in aqueous ammonia; pH 10.0) over 56 min. The flow rate was 200 μL/min, and the eluates were monitored at an absorbance of 214 nm. Thirty fractions were collected and further analyzed using LC–MS/MS.

The samples were separated on a C18 column (75 μm × 25 cm, Thermo Fisher Scientific, Waltham, MA, USA) using a Thermo EASY-nLC 1200 HPLC system (Thermo Fisher Scientific, Waltham, MA, USA). Peptides were eluted and analyzed using a Thermo Scientific^TM^ Q-Exactive Plus mass spectrometer (Thermo Fisher Scientific, Waltham, MA, USA). The separation flow rate was 300 nL/min with a linear 3–100% gradient of buffer B (80% acetonitrile and 0.1% formic acid) over 90 min. The parameters for MS1 were an Orbitrap resolution of 70,000 and an activation type of HCD. The parameters for MS2 were Orbitrap resolution of 17,500 and a dynamic exclusion duration of 30 s. The scan range was 350–1800 m/z, and data collection mode was DDA in top speed.

### 4.5. Database Search and Proteomic Analysis

Protein identification was performed using ProteomeDiscovererTM 2.2 aligned to the NCBInr database (http://www.ncbi.nlm.nih.gov/; accessed on 18 January 2019) and SwissProt/UniProt database (http://www.uniprot.org/; accessed on 18 January 2019) by DIAMOND. Peptide false discovery rate (FDR) analysis ≤0.01 was the filtering parameter. DEPs were identified using a FC (fold change) threshold, and *p*-values of <0.05 were significant. FC > 1.2 was considered upregulated, and FC < 0.83 was considered downregulated.

The COG analysis of the proteins was performed for functional classification via searches in a database (http://www.ncbi.nlm.nih.gov/COG/; accessed on 18 March 2019). Blast2GO 2.5.0 was used for the GO annotations (http://www.geneontology.org/; accessed on 18 March 2019) of the DEPs to classify proteins based on molecular function, biological process, and cellular components. Blast2GO computes Fisher’s Exact Test by applying robust FDR correction for multiple testing and returns a list of significant GO terms ranked by their corrected or one-test *p*-values. KEGG (http://www.genome.jp/kegg/; accessed on 2 April 2020) annotation was performed using KOBAS 2.1.1 software to predict the metabolic pathways. Protein–protein interactions were analyzed using the STRING v10 database (http://strin g-db.org; accessed on 4 February 2021) to determine the identified proteins. A differential metabolic analysis was conducted by Interactive Pathways Explorer (iPath) (http://pathways.embl.de; accessed on 13 April 2020).

### 4.6. qRT-PCR Analysis

The total RNA was extracted from control and TA-treated *S. tuberosum* group Phureja DM1–3 leaves using the Plant Total RNA Extraction kit (Tiangen, Beijing, China) according to the manufacturer’s instructions. cDNA was synthesized from 1 µg of total RNA using the TransScript^®^ One-Step gDNA Removal and cDNA Synthesis SuperMix (Transgene, Beijing, China). qRT-PCR was performed using the SYBR^®^ Green Realtime PCR Master Mix (Toyobo, Osaka, Japan). All qRT-PCRs were analyzed using a CFX96 Touch (Bio-Rad, Hercules, CA, USA) with three technical replicates and three biological replicates. The relative expression levels of selected genes were calculated by the 2^−ΔΔCT^ method [87]. *StActin* (PGSC0003DMG400027746) was used as an internal control. The primers for qRT-PCR are shown in Appendix A.

### 4.7. Histochemical Detection of Hydrogen Peroxide

Hydrogen peroxide detection was carried out using DAB staining, as previously described [88] with minor modifications. Three-week-old potato leaves treated with 10 µM TA and water were tiled on an MS medium in the dark for 24 h. After adding 5 mL of DAB (1 mg/mL), the samples were vacuum-pumped at −0.8 MPa for 5 min and stained on a shaker for 4–5 h at 80–100 rpm in the dark. The staining solution was replaced with 95% ethanol, and the samples were then bathed in boiling water for 10–15 min. Then, the samples were bleached by ethanol (95%) three times.

### 4.8. POD and CAT Activities

Four-week-old potato leaves (2 g) were homogenized in an ice-cold, pre-cooled 50 mM sodium phosphate buffer (pH 7.8)—including 1.0 mM EDTA and 2% (*w*/*v*) polyvinylpyrrolidone—and then centrifuged at 4 °C and 10,000× *g* for 20 min. The supernatant was used for enzymatic assays. The POD activity was detected according to the method of Zhang et al. [89]. Briefly, 100 μL of enzyme extract were mixed with 2 mL of 50 mM phosphate-buffered saline (PBS; pH 7.0), 950 μL of 0.3% H_2_O_2_, and 1 mL of 0.2% guaiacol before being incubated at 37 °C for 3 min. The enzymatic activity was assayed by monitoring the change in absorbance at 470 nm every 1–3 min. One unit of POD activity is expressed as U mg/protein, and 1 unit is equal to the change in absorbance at 0.01 unit per min.

CAT activity was detected using the method of Hanif et al. [90]. The reaction mixture for CAT activity contained 3 mL of 50 mM PBS (pH 7.0), 2 mL of 0.3% H_2_O_2_, and 100 μL of enzyme extract. The change in absorbance of the reaction mixture at 240 nm was recorded every 1–3 min and expressed as U mg/protein. One unit is equal to the change in absorbance at 0.01 unit per min.

## Figures and Tables

**Figure 1 ijms-22-12036-f001:**
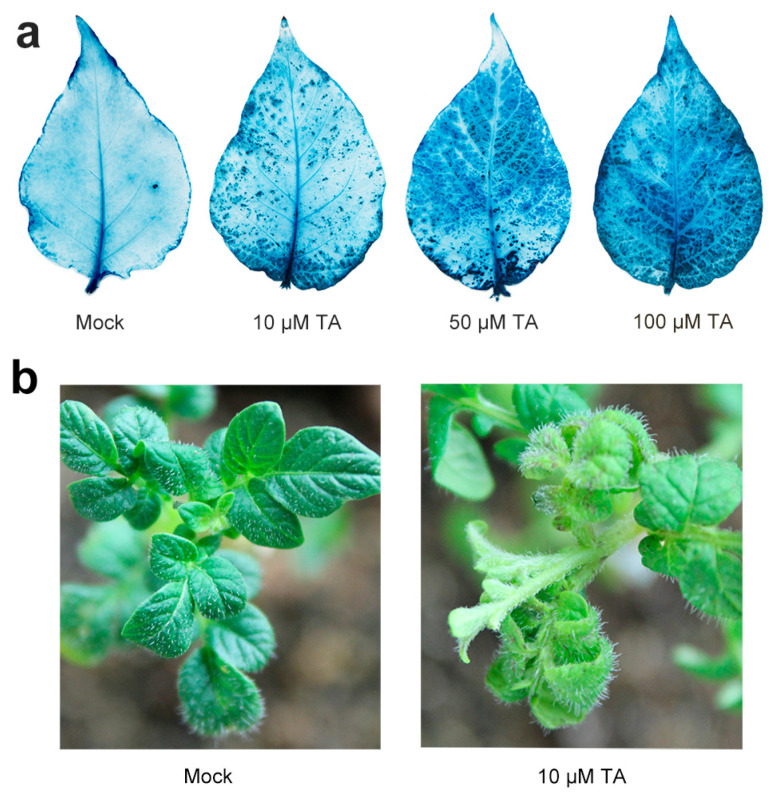
Effects of thaxtomin A (TA) treatment on potato leaves. (**a**) Potato leaves treated with different concentrations of TA for 24 h and stained with trypan blue. The experiments were repeated three times, and similar results were obtained. (**b**) Phenotypes of potato plants treated with sterile water and 10 μM TA for 24 h. All experiments were performed in triplicate with similar results.

**Figure 2 ijms-22-12036-f002:**
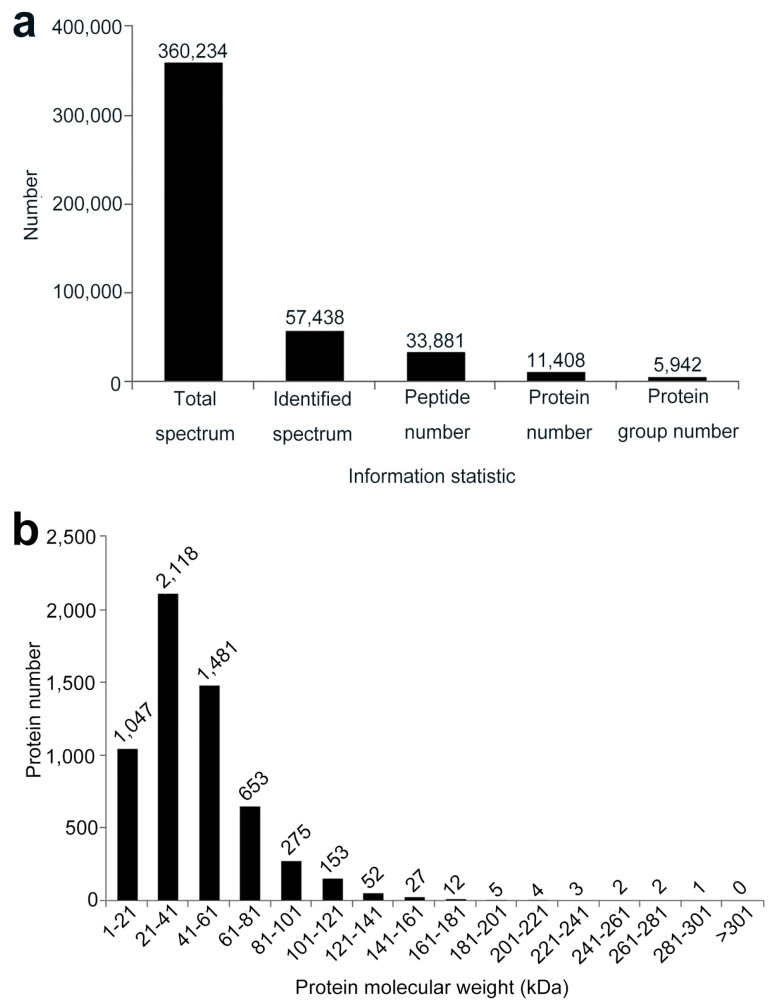
(**a**) Basic protein information statistic for the Isobaric Tags for Relative and Absolute Quantitation (iTRAQ) analysis. (**b**) Distribution of protein molecular weight from the iTRAQ analysis.

**Figure 3 ijms-22-12036-f003:**
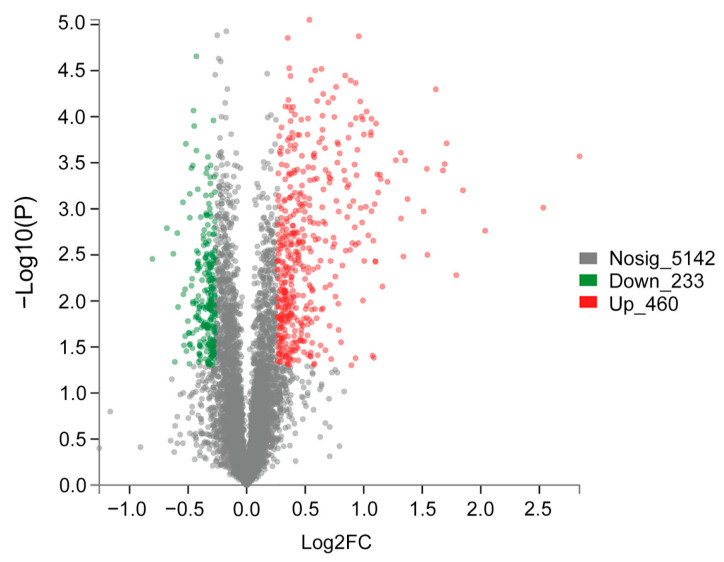
The number of differentially expressed proteins (DEPs) statistics in the form of volcano plots. The red and green dots represent up- and downregulated DEPs, respectively. The gray dots represent insignificant proteins. Significant differences were determined by a Student’s *t*-test.

**Figure 4 ijms-22-12036-f004:**
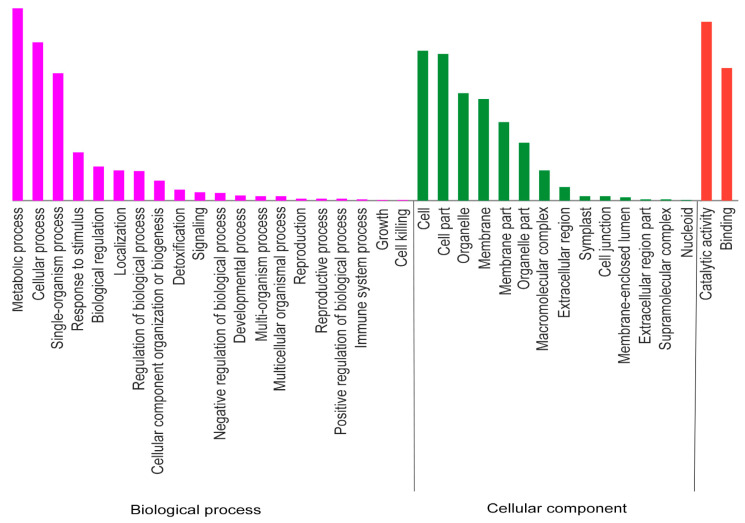
The Gene Ontology (GO) functional annotation of DEPs.

**Figure 5 ijms-22-12036-f005:**
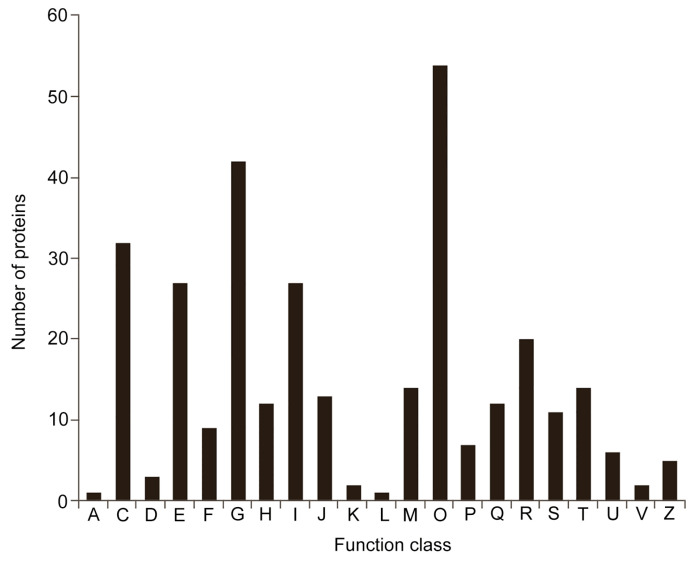
The Clusters of Orthologous Groups (COG) functional annotation of DEPs. A: RNA processing and modification; C: energy production and conversion; D: cell cycle control, cell division, and chromosome partitioning; E: amino acid transport and metabolism; F: nucleotide transport and metabolism; G: carbohydrate transport and metabolism; H: coenzyme transport and metabolism; I: lipid transport and metabolism; J: translation, ribosomal structure, and biogenesis; K: transcription; L: replication, recombination, and repair; M: cell wall/membrane/envelope biogenesis; O: posttranslational modification, protein turnover, and chaperones; P: inorganic ion transport and metabolism; Q: secondary metabolites biosynthesis, transport, and catabolism; R: general function prediction only; R: function unknown; T: signal transduction mechanisms; U: intracellular trafficking, secretion, and vesicular transport; V: defense mechanisms; Z: cytoskeleton.

**Figure 6 ijms-22-12036-f006:**
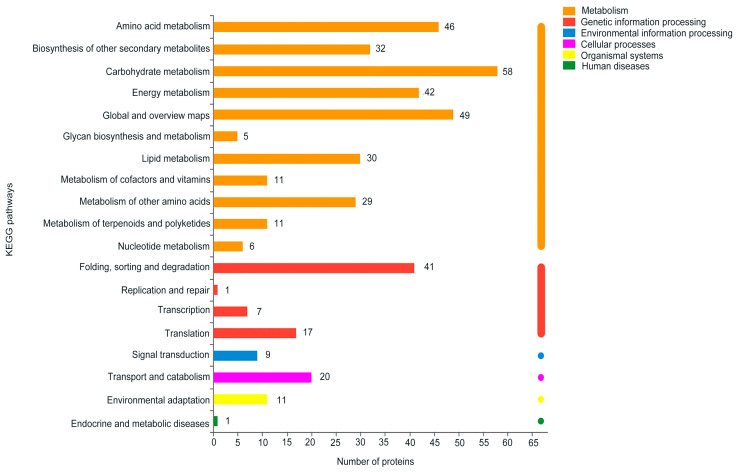
The Kyoto Encyclopedia of Genes and Genomes (KEGG) pathway annotation of DEPs.

**Figure 7 ijms-22-12036-f007:**
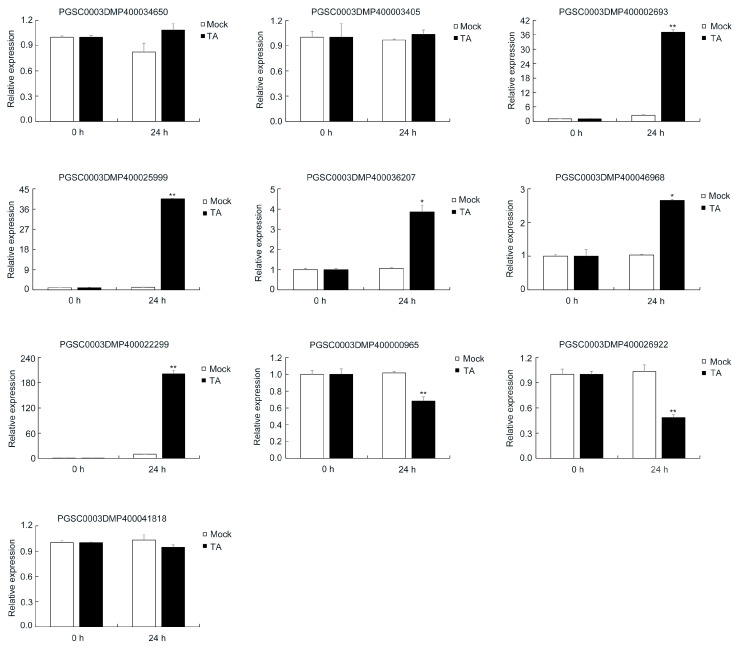
Quantitative reverse transcription-polymerase chain reaction (qRT-PCR) analysis of the expression change correlation between mRNA and DEPs. The asterisks denote statistically significant differences, as determined by a Student’s *t*-test, * *p* < 0.05, ** *p* < 0.01. Three biological repetitions were performed.

**Figure 8 ijms-22-12036-f008:**
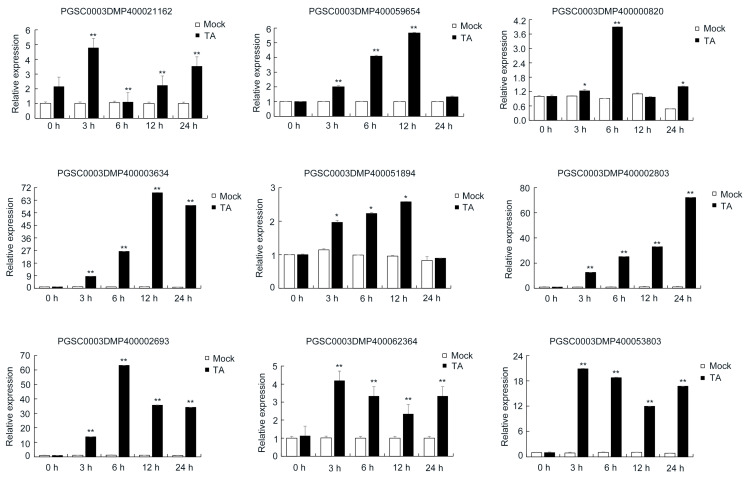
Analysis of the transcript levels of selected DEPs by qRT-PCR. The asterisks denote statistically significant differences, as determined by a Student’s *t*-test, * *p* < 0.05, ** *p* < 0.01. Three biological repetitions were performed.

**Figure 9 ijms-22-12036-f009:**
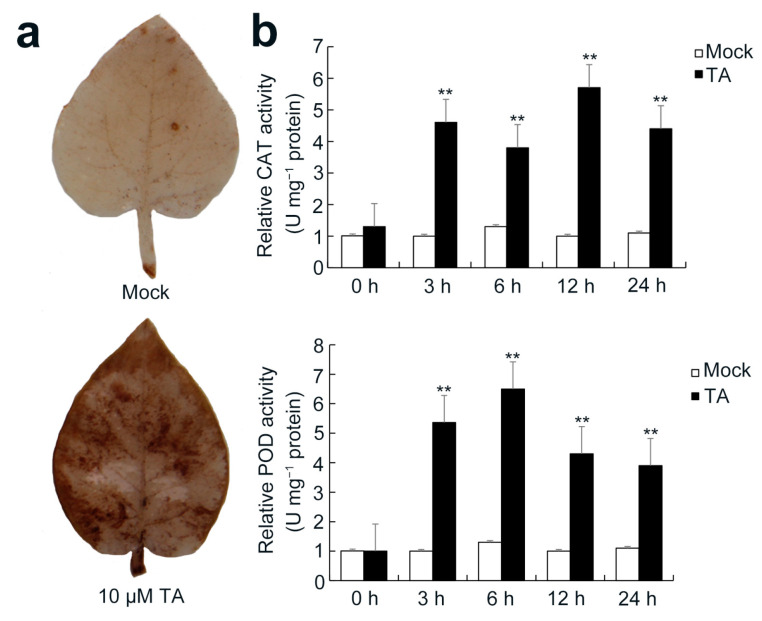
Analyses of 3,3′-diaminobenzidine (DAB) stain and reactive oxygen species (ROS)-related enzymatic activities in TA leaves. (**a**) DAB stain to detect the hydrogen peroxide (H_2_O_2)_ accumulation in leaves; 4-week-old potato leaves were treated with ddH_2_O or 10 μM TA. (**b**) ROS-related peroxidase (POD) and catalase (CAT) activities in leaves. All experiments were performed in triplicate. The asterisks indicate statistically significant differences, as determined by a Student’s *t*-test (** *p* < 0.01).

## Data Availability

The original mass spectra are publicly available from iProX (https://www.iprox.cn/; accessed on 16 September 2021) using the subproject ID IPX0003518001.

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
