# Peer review of "iTRAQ-Based Proteomics Analysis of Response to *Solanum tuberosum* Leaves Treated with the Plant Phytotoxin Thaxtomin A"

_ijms, 2021, doi:10.3390/ijms222112036_

Round 1
Reviewer 1 Report
Dear Authors,
Review of the manuscript entited „iTRAQ-based proteomics analysis of response to Solanum tuberosum leaves treated with the plant phytotoxin thaxtomin A” written by Lu Liu, Liaoyang Hao, Ning Liu, Yonglong Zhao, Naiqin Zhong and Pan Zhao.
In this study, the authors identified 693 DEPs from TA-treated leaves by the iTRAQ technique. Based on the iTRAQ analysis, these DEPs were categorized into different pathways including carbohydrate metabolism, amino acid metabolism, energy metabolism, oxidation-reduction process, response to stress, and secondary metabolic process. Some of the proteomic results were confirmed by qRT-PCR, DAB staining, and enzymatic activity test. The authors found that there is a high correlation between transcript and protein levels, including ROS response, disease and defense response, PR proteins, and secondary metabolites, and also, the few gene transcription level was more significant than protein abundance. TA secreted by Saccharomyces scabies triggered the plant immune response by changes in its metabolism. Since TA can be used to test for resistance to common scab (CS) in potato, so the authors results contribute to the understanding of the response mechanisms of potato plants to TA, and provide a theoretical basis for potato breeding.
I would recommend this interesting manucript for acceptance after minor correction.
My comments are listed below.
A short comment for the Introduction:
lines 41-42: „Thaxtomins are nitro-aromatic compounds that were originally described by Lawrence and King [8,9]. I would suggest to write: „…by King and Laurence [8,9]”.
Results:
line 129:” Then, Go GO functional enrichment analysis were carried out (Table S1)”.
line 179: „Figure 7. qQuantitative reverse transcription-polymerase chain reaction (qRT-PCR) analysis of the expression change correlation between mRNA and DEPs.”
In Figure 8. „Analysis of the transcript level of selected DEPs by qRT-PCR. „ this figure is not in accordance with the text in lines 189-194. E.g. the gene PGSC0003DMT400003774, which codes a defense response gene can not be found on the Fig.8. Furthermore, the genes
PGSC0003DMP400046624 and PGSC0003DMP400003634, respectively, also can not be found on the Figure 8. In the middle row of this Fig.8., there is no nomination for any genes.
Please , correct of these mistakes,namely, harmonize the text and the Figure 8.
lines 209-214: in the Figure 9b when you indicate the CAT and POD activities, you shall have to write at the vertical axis the next: Relative CAT/or POD activity (U mg-1 protein).
line 214: the P letter shall be a big one if you would like to be consistent.
Discussion:
line 248: „, there were 24 DEPs DPEs in the biosynthesis of amino acids pathway; other DPEsDEPs….”
Mat&Methods:
lines 357- 358:” propagated on Murashige and Skoog (MS) medium containing 30 g/L sucrose and 0.8% agar []”. The proper citation is missing.
line 375: TA was quantified by high-performance liquid chromatography (HPLC). The specification of HPLC is missing: firm, city, state where it is arised from.
line 380: „The necrotic spots in the leaves were observed by trypan blue stain [].” Citation for the staining method is necessary.
Two questions: why did you use two Solanum, one (S. tuberosum cultivar Shepody) for TA test and another (S. tuberosum group Phureja DM1-3) for the iTRAQ or qRT-PCR analysis? Did the S. tuberosum group Phureja DM1-3 plants also show the signs of PCD after TA treatments?
lines 418- 419: „using a Thermo Scientific Vanquish Flex Ultra-High Performance Liquid Chromatography (UHPLC) system equipped with an Acquity UPLC BEH C18 column (2.1 × 150 mm, 1.7 μm) (firm, city, state is missing).
lines 424, 425, 427: the specifications are missing for Thermo Fischer Scientific - city, state is missing
line 434: „….the NCBInr database and SwissProt/UniProt database by DIAMOND” (website or other specification is missing)
line 449: „…DM1-3 leaves using the Plant Total RNA Extraction kit (Tiangen, DP432) according to..” – here also the specification (city, state) is missing
line 465:„ samples were bleached three times , and images were obtained „ by what???
Sincerely yours,
Reviewer 1
Reviewer 2 Report
In this manuscript, the authors have used iTRAQ and qPCR analyses to study differentially accumulated proteins in response to treating Thaxtomin A (TA) in potatoes.
While the study is exciting and the manuscript is very wee-written, some essential issues should be highlighted:
For global differential transcriptome and proteomics analyses, scientists usually use Fishers-test/T-test to find the differentially GO group (Figure 6).
In my humble opinion, qPCR analysis is not necessary since the differentially accumulated proteins are the primary target of this study. As proteins are the final stage in biological dogma, authors should not go back and check the transcript levels. A transcript could be produced in the cell and may not be translated to a functional protein, but this could not be in the opposite direction. Consistency between protein and RNA levels is not an issue in this case because you are not here evaluating the iTRAQ V.S. qPCR systems. Instead, the authors could confirm the global proteomics results by targeting specific proteins using western blot or immunohistochemistry analysis.
Therefore, I suggest that the author confirm the proteomics results using specific antibodies.
Reviewer 3 Report
Title: iTRAQ-based proteomics analysis of response to Solanum tu-2 berosum leaves treated with the plant phytotoxin thaxtomin A
Authors: Lu Liu, Liaoyang Hao, Ning Liu, Yonglong Zhao, Naiqin Zhong, and Pan Zhao.
Journal: International Journal of Molecular Science
Manuscript number: ijms-1405429
General remarks: The manuscript by Liu et al. reported an investigation about the proteomics effect of the phytotoxin thaxtomin A (TA) on Potato leaves. The approach is interesting and proteomics would be an effective strategy the simulate pathogen attack and to identified the effects of TA on Potato. Techniques as transcriptomics and/or proteomics still remain powerful and attractive resources for scientist. Unfortunately in the present version of this manuscript, the authors limited themselves in listing the most differential expressed proteins and perform a gene onthology enrichment analysis. This is not sufficient to complete this type of analysis and analyze the effects of the selected treatment. These types of data should be deeply analyzed and compared with other analysis to obtain clear and novel hypothesis. Beyond this personal opinion, the present version of the manuscript showed a number of lacking points. Particularly this is very poorly written reporting a number of elementary grammar errors. The English language is not adequate for a scientific paper. Finally, the discussion should be improved. A number of discussion points argued by the authors such as the regulation of aminoacids biosynthesis and the ROS accumulation in plants subjected to TA treatments were previously reported in other papers. Authors should improve the novelty of their results deeply analyzing their new outcomes. Based on these considerations this paper is rejected for publication on International Journal of Molecular Science. Considering the value of the data, I suggest the authors to take an adequate period to revise and improve the manuscript and then resubmit it.
Round 2
Reviewer 2 Report
I think authors should validate their findings using western blotting.
Reviewer 3 Report
The present version of the manuscript by Liu et al. is suitable for publication on IJMS.
Please pay attention to line 92. The authors probably confused potato with tomato. Please revise.
